# Zebrafish Embryos Display Characteristic Bioelectric Signals during Early Development

**DOI:** 10.3390/cells11223586

**Published:** 2022-11-12

**Authors:** Martin R. Silic, Ziyu Dong, Yueyi Chen, Adam Kimbrough, Guangjun Zhang

**Affiliations:** 1Department of Comparative Pathobiology, Purdue University, 725 Harrison Street, West Lafayette, IN 47907, USA; 2Department of Basic Medical Sciences, Purdue University, 625 Harrison Street, West Lafayette, IN 47907, USA; 3Weldon School of Biomedical Engineering, Purdue University, 206 South Martin Jischke Drive, West Lafayette, IN 47907, USA; 4Purdue Institute for Inflammation, Immunology, and Infectious Diseases (PI4D), Purdue University, 298 Nimitz Dr, West Lafayette, IN 47906, USA; 5Purdue Institute for Integrative Neuroscience, Purdue University, 207 South Martin Jischke Drive, West Lafayette, IN 47907, USA; 6Purdue Center for Cancer Research, Purdue University, 201 S. University Street, West Lafayette, IN 47907, USA

**Keywords:** bioelectricity, embryogenesis, development, zebrafish, cleavage, blastula, gastrulation, somite, ASAP1, cellular membrane potential, Vm

## Abstract

Bioelectricity is defined as endogenous electrical signaling mediated by the dynamic distribution of charged molecules. Recently, increasing evidence has revealed that cellular bioelectric signaling is critical for regulating embryonic development, regeneration, and congenital diseases. However, systematic real-time *in vivo* dynamic electrical activity monitoring of whole organisms has been limited, mainly due to the lack of a suitable model system and voltage measurement tools for *in vivo* biology. Here, we addressed this gap by utilizing a genetically stable zebrafish line, Tg (*ubiquitin*: ASAP1), and ASAP1 (Accelerated sensor of action potentials 1), a genetically encoded voltage indicator (GEVI). With light-sheet microscopy, we systematically investigated cell membrane potential (Vm) signals during different embryonic stages. We found cells of zebrafish embryos showed local membrane hyperpolarization at the cleavage furrows during the cleavage period of embryogenesis. This signal appeared before cytokinesis and fluctuated as it progressed. In contrast, whole-cell transient hyperpolarization was observed during the blastula and gastrula stages. These signals were generally limited to the superficial blastomere, but they could be detected within the deeper cells during the gastrulation period. Moreover, the zebrafish embryos exhibit tissue-level cell Vm signals during the segmentation period. Middle-aged somites had strong and dynamic Vm fluctuations starting at about the 12-somite stage. These embryonic stage-specific characteristic cellular bioelectric signals suggest that they might play a diverse role in zebrafish embryogenesis that could underlie human congenital diseases.

## 1. Introduction

All living cells have a membrane potential (Vm), making bioelectricity an essential property of life. Bioelectricity is endogenous electrical signaling mediated by the dynamic distribution of charged molecules [1,2,3,4]. The importance of bioelectric regulation has been shown in various fields such as neuromuscular, embryogenesis, cancer, wound healing, regeneration, tissue patterning, and cell migration [5,6,7,8]. The critical functions of electrical signaling during early embryonic development have been proposed for years, mainly based on indirect results. Mutations in a variety of ion channels and other regulators of charged molecules have been shown to cause a vast range of phenotypes, such as alterations to normal limb formation, craniofacial malformations, as well as heart and neurological disorders in multiple distinct species [9,10,11,12]. For example, the injection of *KCNA5* mRNA into *Xenopus* embryos induced the growth of ectopic eyes [13]. In addition, we recently found that transient ectopic expression of *kcnj13* in the somites could cause a long-finned phenotype in adult zebrafish [14]. Furthermore, changes to channels and gap junctions can alter normal pigment patterning [15,16,17,18,19]. All these results point to bioelectric signals playing an essential role in normal embryonic development. However, systematic real-time direct evidence of bioelectricity during vertebrate embryonic development has been lacking. Although, electrochemical dyes and electric probes in *Xenopus* embryos give some indications of the role bioelectricity plays in embryonic development [20,21]. The main reasons for this lack of data are the limitations of the model system and voltage measurement tools for *in vivo* biology.

Zebrafish embryos are a superior system for studying developmental biology due to many advantages such as rapid external development, transparency of early embryos, and tractable genetics [22,23]. The stages of zebrafish embryogenesis have been well characterized. Females and males release their gametes into the water, where oocytes are fertilized and begin a synchronous meroblastic cleavage process. They are classified as discoidal, where the group of dividing cells sit atop a large yolk and eventually form the blastula. This ball of cells continues to multiply and eventually migrates down the yolk to form the three germ layers during gastrulation. The early embryo transparency and ease of genetic manipulation make zebrafish an ideal model for vertebrate imaging studies, and much progress has already been made in many research fields such as neuroscience and organogenesis [24,25,26,27,28].

With advances in modern neuroscience, genetically encoded biosensors have been developed to overcome the limitations of chemical dyes [29]. Genetically encoded biosensors generally allow sensitive and real-time dynamic assays for monitoring cells under natural physiological conditions. While chemical dyes/probes usually have limited lifetimes, relatively slow response, and delivery challenges due to tissue specificity and penetration. Thus, the use of GECIs (genetically encoded calcium indicators) and GEVIs (genetically encoded voltage indicators) has increased in recent years. These tools have already been applied successfully in many model systems for monitoring real-time dynamic bioelectric signals *in vivo* [30]. Moreover, both types of genetically encoded indicators have also been validated in zebrafish embryos [31,32,33,34]. For example, GCaMP6s provided an excellent temporal and spatial resolution of calcium signaling during zebrafish embryogenesis, and revealed previously missed signal information not visible with dyes [33,34]. One of the commonly used GEVIs, ASAP1 (Accelerated Sensor of Action Potentials 1), has also been effective at reporting zebrafish neuronal activities within developing embryos [31,32]. Thus, these newly developed GEVIs and improved fluorescent imaging tools such as light sheet microscopy (LSM) provide an unprecedented opportunity to measure endogenous bioelectricity with enhanced sensitivity, signal-to-noise, acquisition speed, kinetics, and reduced toxicity and tissue damage [35,36].

In this work, we took advantage of our Tg (*ubi*: ASAP1) transgenic zebrafish and systematically analyzed endogenous bioelectric signals in early zebrafish embryos using LSM. To our knowledge, this is the first real-time systematic analysis of endogenous bioelectric signals during vertebrate embryonic development. We found zebrafish embryos show characteristic bioelectric signals at corresponding embryonic developmental stages, suggesting their versatile functions.

## 2. Results

We have generated a Tg (*ubi*: ASAP1) transgenic fish line that can report endogenous bioelectric signals. The *ubi/ubiquitin* promoter lines allow for expression in all cells during embryogenesis, and the fluorescent signal can be visualized using an epifluorescence microscope [31]. However, we have not systematically investigated the electric signal due to the relatively low fluorescence intensity, high signal speed, and phototoxicity. To record these changes with sufficient speed and reduced tissue damage, we turned to LSM (Figure 1A–C), which overcomes the challenges presented by this type of imaging with epi-fluorescent microscopy.

### 2.1. Cleavage Furrow Hyperpolarization Precedes Cytokinesis and Becomes More Dynamic as Zebrafish Embryos Develop in the Cleavage Period

An intriguing phenomenon we have noticed in Tg (*ubi*: ASAP1) fish embryos, is the local cell membrane hyperpolarization during the cleavage stage (Figure 2, Appendix A). To better understand and quantify this hyperpolarization, we examined the Vm signal of cleavage-stage embryos using a high-speed LSM. Cell membrane voltage can be detected even in unfertilized embryos, which showed randomly positioned signals and variable shapes of Vm fluctuations (Appendix A). In fertilized 1-cell stage fish embryos, we first observed ASAP1 signals (brighter fluorescence) localized to the initial cleavage plane before the cell was cleaved in half (Figure 2A–G). The initial “center furrow” from the first cleavage of the 1-cell stage remained hyperpolarized (Figure 2H), and this dynamic signal persisted into the subsequent cell division. Meanwhile, the 2-cell stage embryo began to show a hyperpolarization signal at the center of each cell (parental cells, P1 and P2) (Figure 2H–N, Appendix A). One of the parental cells, P1, showed stronger signaling throughout the division. This signal started in the middle of each cell perpendicular to the first division plane and moved bi-directionally outward. The cell membrane of the P2 cell showed a similar bioelectric signal to P1, which could potentially be linked with cleavage furrow positioning and propagation. To better understand these signals, we defined regions of interest (ROIs) to calculate changes in fluorescence intensity at the locations of the furrows over time. Indeed, the furrows of the 2–4 cell stage transition showed that the fluorescent change (∆F) in P1 was the strongest overall, with P2 following a weaker change (Figure 2CC). The center furrow also displayed Vm changes while the two new furrows formed. Noticeably, all furrow-related hyperpolarized signals did not remain stable, as fluctuations were clearly noticed as cytokinesis progressed. (Figure 2I–N,CC, Appendix A).

The 4-cell stage embryos had signals remaining at P2 furrows (Figure 2O, white arrow) before new signals appeared at the center of the newly dividing cells. All four cells showed different initial fluorescence intensities (Figure 2O–U). By our ROI quantifications, the furrows of the 4–8 cell stage transition showed a similar pattern to the 2–4 cell stage divisions (Figure 2CC,DD, Appendix A). The remaining signals from the previous furrows were stronger before the new divisions (P1, P2, Figure 2DD, Appendix A), but gradually decreased before the new furrows formed. The initial four peaks of DC1-4 matched well but became less synchronized as cytokinesis progressed (Figure 2P–U). In most embryos we imaged (n = 8 out of 9), the left daughter cells (DC1 or 2) showed signals first (Figure 2P), then the right daughter cells, (DC3 or 4) (Figure 2Q, Appendix A). However, this observation is not always consistent. One fish embryo showed a diagonal pattern (DC1 to DC3) (Appendix A). The cleavage furrow hyperpolarization signals continued in a comparable way for the 8–16 and 16–32 cell stages. However, the initial signal timing and intensity difference were more variable than in the 4-cell stage. Starting at 8–16 cells, less synchronized and more dynamic oscillations occurred at the furrows of newly dividing cells (Figure 2V–BB, Appendix A).

### 2.2. Whole-Cell Vm Transient Signals Are Located in the Superficial Blastomere during the Zebrafish Blastula Period

As zebrafish embryos develop into the blastula stage, cell number increases, but cell volumes decrease due to discoidal cleavage. With max intensity projections of Z-stack timelapse videos, we found that the electric signal mainly exhibits whole-cell Vm transients instead of cleavage-furrows membrane local signal (Figure 3A–L, Appendix A). Interestingly, most whole cell Vm transients (Figure 3A,B) were distributed over the embryo surface of the enveloping layer (EVL) as well as the yolk syncytial layer (YSL). Individual cells (in multiple frames) showed a dynamic nature of electric signals during this embryonic period (Figure 3A–C,H–L). To further detect and track these signals, we turned to Oxford Instruments Imaris software (9.7.2 Bitplane AG) for signaling analysis. With time-lapse videos (Total time 30 min, 5-s intervals between Z-stacks), we were able to count the number of Vm transients over time and calculate the duration of transients. Embryos (n ≥ 5) were either classified as “early” (2.5–3.5 h. or 512 cells to high stage) or “late” (3.5–4.5 h. or oblong to dome) blastula stage. Imaging analysis of the early blastula stage revealed that transient numbers fluctuated over time, with periods of a higher and lower number of signals in each frame (Figure 3M). We then turned to the tracking feature in the Imaris program, which allowed one transient event to be counted once, even if the same cell displayed bright fluorescence in multiple frames. We found that more Vm transients were occurring in the early blastula (~727) compared to the later blastula period (~284) (Figure 3N). The average transient duration (about 10 s) did not differ much between the two blastula stages (Figure 3O). To examine whether these signals were within the deeper cells, we examined a single plane Z-slice and found that the signals were limited to the outer edge of the blastomere with a lateral slice from both the lateral position (Figure 3P, Appendix A) and from the view of animal pole (Figure 3Q). The superficial blastomere signaling was observed in both the early and late blastula stages. Intriguingly, we observed sequential Vm signaling occurrences between adjacent cells (Figure 3R–AA, Appendix A), suggesting that Vm could function as an intercellular signal.

### 2.3. Whole-Cell Vm Transient Signals Occur More Frequently during the Zebrafish Gastrula Period but with Similar Signal Duration

When the fish embryos develop to the gastrula period, we chose imaging with longer total times and intervals to capture an overall picture of Vm dynamics during this stage. Time-lapse imaging revealed a continuation of Vm transients within the early stages of gastrulation (4.5–6 h or 30% epiboly to shield) and within the later stages (6–8 h or shield to 75% epiboly) (Figure 4A–AA, Appendix A). Early gastrulation period Vm transients frequently fluctuated as in the blastula period. However, the number of Vm transients increased without the Vm transient duration being significantly affected (Figure 4G–I vs. Figure 3M–O). Since Vm transients were only observed within the EVL (enveloping layer) and YSL (yolk syncytial layer) during the blastula stage, we decided to check if this held true during the gastrula period, in which the three germ layers are formed by dynamic cell movements and internalizations. Indeed, we found signaling within the deep cells during the gastrula period, starting at around 30% epiboly (Appendix A). We could also detect Vm signals occurring within layers deeper than the superficial blastomere (Figure 4P–U, Appendix A).

### 2.4. During the Segmentation Period, There Are Tissue-Level Dynamic Cellular Bioelectric Signals

When the fish embryos moved into the segmentation period, sporadic transient electric signals continued to occur all over the embryo. However, more tissue-level changes began to occur. Certain regions, such as the somites, became more hyperpolarized than surrounding tissues (Figure 5A–F, Appendix A). The Vm signals in some other tissues, such as the developing heart, also showed more obvious electrical signal fluctuations (Figure 5J–K). At about the 12-somite stage, Middle-aged somites became strongly hyperpolarized (Figure 5G–L, Appendix A). Interestingly, the somite signal was also dynamic, occurring in whole or partial somites. In addition, either unilateral or bilateral somites showed strong hyperpolarization (Figure 5M–R, Appendix A). To quantify these somite signals, we divided the embryo trunk into seven ROIs, starting at the middle of the trunk along the dorsal side down to the tailbud region (Figure 5S). Mean fluorescence intensity changes over time were tracked, and ∆F was calculated. As development progressed, we found that somite region fluorescence intensity gradually increased as the embryos further developed. Moreover, middle to posterior somite regions, such as ROI-4 and ROI-5, showed a greater amount of signaling events (Figure 5T,U). In contrast, the first few anterior somites did not show many signal fluctuations at this stage (Figure 5T,U). There was a significant difference between the anterior and posterior somites and even significant differences among the other middle regions.

## 3. Discussion

Mounting evidence has suggested that bioelectric signaling plays a significant role in embryonic development. However, direct evidence of embryonic bioelectric signaling has not been yet available. Here, we revealed that zebrafish embryos display characteristic bioelectric signals at corresponding embryonic developmental stages (Figure 6) using newly developed technologies such as GEVI and LSM. These results lay the fundamental groundwork for understanding the endogenous electrical signaling patterns accompanying the initial stages of zebrafish embryonic development.

Our study revealed that bioelectric signals are present even within unfertilized embryos and within the initial cleavage plane of the 1-cell stage embryos. Cell membrane hyperpolarization around the cleavage furrow preceded and persisted during the early divisions in a highly dynamic fashion. Moreover, the cleavage furrow signal continued but fluctuated when cytokinesis progressed due to the dynamic process of cytokinesis and the incomplete meroblastic cleavage of zebrafish embryos. Overall, bioelectric signals of this stage remained localized to the furrows and tended to be slightly asynchronous among the newly formed cells. However, we did notice the initial furrow signal could appear within cells on one side of the embryo or cells first appearing diagonally to one another during the 2-to-4 stage transition. However, this scenario was much less frequently observed.

In contrast, the bioelectric signals transitioned to whole-cell Vm transient events once fish embryos reached the blastula period. We found that the Vm transients concentrated in the superficial regions, EVL and YSL, where cell divisions frequently occurred. This suggests that the signal could still be related to cell divisions. Interestingly, we also found intercellular sequential transients, which indicated that electric signaling might also be utilized for tissue-level communication. During the gastrulation period, Vm transients remained dominant in the margin of the embryos. However, they began to show in the deeper cells at about 30% epiboly. Compared to early gastrulation, the Vm transient number decreased in the later stages of gastrulation but not the bioelectric transient duration. This could be due to missed signals because we utilized a lateral position. Only one side of the embryonic cells was captured. Conversely, imaging from the anterior-posterior view would not detect the signaling of migrating cells down the sides of the yolk. Therefore, imaging half of the embryo might mean the total number of transients at this stage would be roughly doubled.

During the late gastrulation and segment periods, tissue-level hyperpolarization was observed in somites. These tissue-level bioelectric signals may be correlated to tissue differentiation. As the fish embryos marched into the segmental period, strong somite-level bioelectric signals became more dynamic, supporting the idea that they are related to tissue patterning and differentiation. All these characteristic bioelectric signals corresponded to specific embryonic developmental stages, indicating their intrinsic roles. However, the underlying ion channels and connexins that generate these signals are unknown. Our recent gene expression analysis of calcium-gated potassium channels (KCa) and inwardly rectifying potassium channels (Kir) revealed that many (*kcnn1b*, *kcnn3*, *kcnma1a*, *kcnma1b*, *kcnmb2b*, *kcnmb3*, *kcnj4*, *kcnj2a*, *kcnj2b*, *kcnj11*, *kcnj5*, *kcnj21*) have a somite-specific expression at similar developmental stages [37,38]. Their presence in the developing somites may indicate that these channel activities underlie the tissue-level bioelectricity. Future experiments on disrupting these potassium channels by CRISPR may prove their contribution to somite bioelectrical signaling. Another interesting phenomenon we noticed is that neural tissues did not show more electric activities than other tissues in early zebrafish embryos, especially the newly formed somites. As the embryos are not mobile at this stage, it is unlikely that these strong Vm changes are due to movement. Instead, this may indicate the bioelectric signal could be crucial to somite differentiation, such as epithelial-to-mesenchymal transition and dermomyotome differentiation. Perturbation of such electric signals may have a dramatic impact on adult zebrafish body patterns. For example, the long-fin fish Dhi2059 mutant was caused by an ectopic expression of *kcnj13* in the somites [14]. It is also interesting to note that the location of ectopic *kcnj13* expression in Dhi2059 mutant fish during the somite stage is within the Middle-aged somites. Coincidently, this is the same tissue where our ASAP1 reporter line showed the most electrical activity.

The functions of these unique developmental stage-specific bioelectric signal patterns during zebrafish embryogenesis remain largely unexplored. They could be related to cell cycle or cytokinesis, as previously suggested by ion channel studies from multiple species [39,40]. As most Vm transients were found in the peripheral regions during the blastula and gastrula stages, they likely play instructional roles in cell growth, differentiation, and organ patterning. As electric signals are correlated with calcium signals in neural tissues, it is also possible that the electric transients are just a reflection of calcium signal alterations in certain tissues, although this possibility is not high. Another possibility is the opposite, the electric signals trigger calcium signals.

In the field of neuroscience, calcium signals have been used as a surrogate marker of neuronal firing and electrical activity, and recent comparative studies have confirmed the two have a good correlation [41,42,43]. Calcium signaling has been extensively investigated in zebrafish embryos [33,44,45,46,47]. Our observations of bioelectric signals share many similarities with reported calcium signals. Both are correlated to embryonic developmental stages from cleavage furrow localized patterns to whole-cell transients and intercellular occurrences [44]. These similarities suggest both might be involved in similar biological functions during embryogenesis. It is worth noting that single-cell organisms such as bacteria and protozoans, without a nervous system, still have calcium signaling and electrical activity, evidenced by the presence of ion channels, Vm, and even neurotransmitter activity [48,49]. Thus, bioelectricity and calcium, as important regulators, may have evolved before the development of neural tissue in these species. In addition to similarities, we did notice differences between the two types of signals. When compared to previously reported calcium signaling by GCaMP6G in zebrafish embryos, we find that transient Vm signals are more numerous and occur more rapidly. This may indicate that the Vm reporter could be more sensitive than the calcium one, due to its nature as a secondary messenger [50,51]. However, these differences also could be caused by the slower imaging speed in the GCaMP6Gs study [33]. Similarly, it is also difficult to directly compare our data with previously reported studies with calcium dyes [44,45,46].

In summary, this report revealed early zebrafish embryos’ first real-time endogenous bioelectric signals. Future investigations with improved GEVIs and genetic tools will expand our understanding of bioelectricity, especially its relationships with traditional developmental signaling pathways such as morphogen proteins (e.g., WNT) and transcriptional factors (e.g., HOX) [52,53]. In the future, the biological roles of embryonic Vm could be further examined with zebrafish ion channel mutants, newly developed optogenetic, or chemogenetic tools such as DREADDs (Designer Receptors Exclusively Activated by Designer Drugs) or uPSAM (ultrapotent Pharmacologically Selective Actuator Modules) [54,55,56].

## 4. Materials and Methods

### 4.1. Zebrafish Strains and Transgenic Fish Line Husbandry

Zebrafish were raised and maintained within the Purdue veterinary hospital animal housing facility (West Lafayette, IN. USA), which was approved by the Association for Assessment and Accreditation of Laboratory Animal Care (AAALAC). Purdue Animal Care and Use Committee (PACUC) approved protocols were used to perform experiments. All zebrafish trials were conducted in wild-type TAB fish genetic backgrounds. Zebrafish were maintained according to the zebrafish book, and embryos were staged according to the Kimmel staging guide [23]. The Tg (*ubi*-ASAP1) fish line was generated in our previous report [31].

### 4.2. Imaging Early Zebrafish Embryo Vm Fluorescence and Data Analysis

Multiple Tg (*ubi*: ASAP1) adult fish were in-crossed or out-crossed with TAB fish to acquire green fluorescence-positive offspring. Zebrafish embryos were collected at different desired developmental stages. To better visualize the cellular GEVI-GFP activity, zebrafish embryo chorions were either left in place or carefully removed using a pair of forceps under a dissection scope before mounting in 0.6% low melting agarose (IBI Scientific CAS#9012-36-6) on a sample platform to maintain their positions.

Zebrafish embryos were imaged using a Miltenyi Biotec light sheet microscope, UltraMicroscope II with a Super Plan Module configuration, a 4x NA 0.35 MI PLAN objective, and ImspectorPro software (7.1.4 Lavision Biotec, Bielefeld, Germany). Image acquisition total times varied from minutes to 16+ hours depending on embryonic stages. Z-stacks between 1 and 20 slices had total intervals between 0.5 s and 3 min. Laser power was set between 50–70% and sheet width at 60% for image acquisition. Water was selected as the imaging medium. Exposure times were between 50 ms and 300 ms, depending on the imaging speed.

Max intensity projections were used to display 3D images by importing TIF files to ImageJ [57]. ROIs were placed over areas of embryos with signals to track mean fluorescence changes over time. Fluorescence intensity data were exported into Excel for further analysis. The ∆F_Adj_ was calculated as (F_t_ − F_0Adj_)/F_0Adj_, where F_t_ is the fluorescent value at a given time t, and F_0_ is the baseline fluorescence constant value. F_0Adj_ was calculated by averaging at least four frames without any bright GFP signal. Traditional ∆F/F was also calculated in the Appendix A as (F_t_ − F_0_)/F_0,_ where F_0_ is equal to F_t (n−1)_. Vm transient signals were analyzed using Imaris software (9.7.2 Bitplane AG). Time-lapse Imaging files were converted to .ims format and imported to the Imaris program. The “spots” function was used to detect electric transient fluorescent signals within an ROI of a given embryo (n ≥ 5). For the algorithm, default parameters were used. Estimated XY diameter was based on cell diameter measured within the Imaris slice tab and generally fell between 10–20 μm depending on the embryonic stage. Background subtraction was selected. The signal “quality” parameter in Imaris for detection was set at a sufficient “level” using the slide bar, which could detect transient signals without recognizing background noise, generally between 80 and 100+. The “Tracks” function was used to determine the total number of Vm transients over time so that a signal was counted only one time if appearing in multiple frames to define transient number and duration. Autoregressive motion, an algorithm that allows for tracking back an immediately previous time point, was selected with “Max Distance” set as a value equal to the diameter of the cell and a gap distance of zero. After completing the analysis, data was converted and saved into an Excel file format. “Track Duration” statistics gave the total number of transients and the different transient durations. GraphPad Prism (v9.4.1) was used to generate graphs and perform statistical calculations. The student’s *t*-test was used to determine the statistical significance between groups.

## Figures and Tables

**Figure 1 cells-11-03586-f001:**
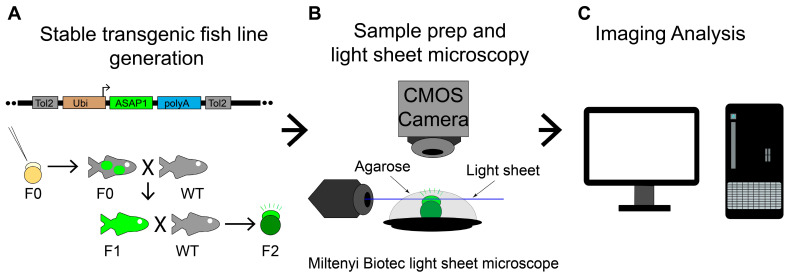
Overall experimental procedure. (**A**) Illustration of a Tol2 construct, the method to produce stable ASAP1 zebrafish line, Tg (*ubi*: ASAP1), and zebrafish crosses. X indicates fish cross. Black arrows show fish raising or producing. The green color labels the ASAP1 transgene. (**B**) Experimental setup to image zebrafish Tg (*ubi*: ASAP1) embryos with Miltenyi Biotec light sheet ultramicroscope II. ASAP1-positive embryos were mounted in agarose on a platform to keep them stable during imaging. (**C**) Image analysis was performed using ImageJ (v1.53e) and Imaris programs (9.7.2, Bitplane AG).

**Figure 2 cells-11-03586-f002:**
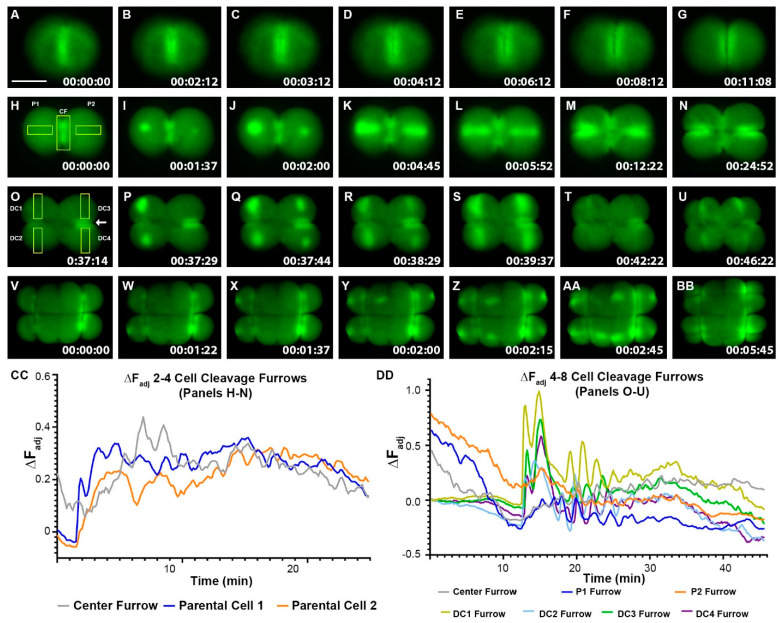
Zebrafish Cleavage period embryos display furrow-related dynamic hyperpolarization. (**A**–**BB**) Still-frame representative max-projection images from time-lapse videos (Appendix A). 1–16 cell stages of Tg (*ubi*: ASAP1) zebrafish embryos were imaged from the animal pole position. (**A**–**G**) Representative Vm images from 1–2 cell stage fish embryo. (**H**–**U**) Representative Vm images from 2–8 cell stage fish embryo. (**V**–**BB**) Representative Vm images from an 8–16 cell stage fish embryo. Areas of bright green indicate hyperpolarization. Yellow boxes show regions of interest (ROIs) for measuring fluorescence intensity over time. The white arrow in (**O**) points to the P2 furrow signal. Signals appeared before cleavage furrows formed and then fluctuated as cytokinesis progressed. (**CC**) Adjusted fluorescence intensity, ∆F_Adj_, of ROIs in panels (**H**–**N**). (**DD**) Adjusted fluorescence intensity, ∆F_Adj_, of ROIs in panels (**O**–**U**). All lines in panels (**CC**,**DD**) represent the change in adjusted fluorescence intensity of ROIs for the designated cleavage furrows over time. *CF* (center furrow), a fertilized embryo’s initial division plane. *P1*, parental cell one. *P2*, parental cell two. *DC1*, daughter cell one. *DC2*, daughter cell two. *DC3*, daughter cell three. *DC4*, daughter cell four. Time (lower right corner), hours: minutes: seconds. Scale Bar= 250 µm.

**Figure 3 cells-11-03586-f003:**
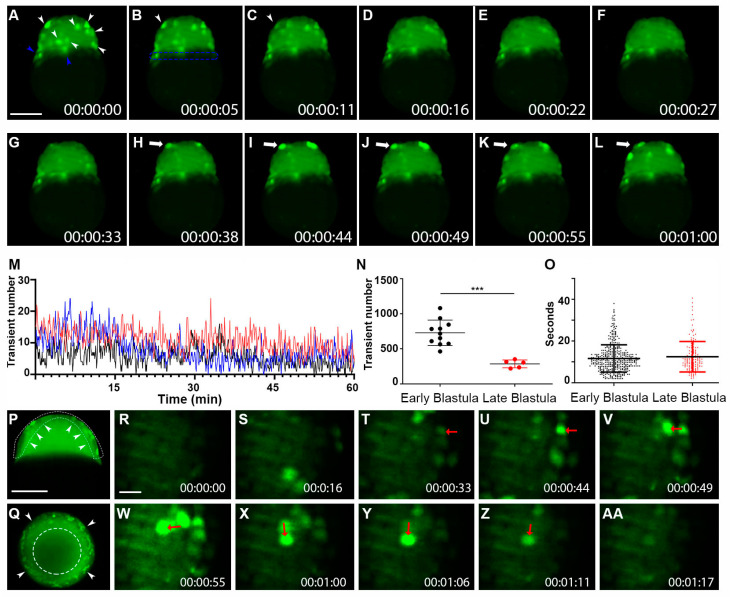
Whole-cell Vm transients occurred in the zebrafish superficial blastomeres during the blastula period. (**A**–**L**) Still-frame representative max-projection images from a time-lapse video (18 min total time, 5.5-s intervals, Appendix A). Early-stage blastula of the Tg (*ubi*: ASAP1) zebrafish embryo was imaged from a lateral position. (**A**) White arrowheads indicate whole cells that were hyperpolarized. Blue arrowheads point to Vm signals in YSL. (**B**) The blue dashed line indicates the YSL region of cells. Arrowheads in panels (**B**,**C**) show the same cell with signal fading over time. (**H**–**L**) White arrows show a cell that became hyperpolarized and eventually faded after about 20 s. (**M**) Average number of transients occurred at a given time point from a 60 min acquisition. The total number of hyperpolarized cells fluctuated over time. Each colored line indicates different fish embryos. (**N**) The total number of Vm transients occurred within the early (2.5–3.5 h) and the late (3.5–4.5 h) blastula (n ≥ 5 embryos for each group). Asterisks indicate a statistical significance of *p* < 0.001. (**O**) Vm transient duration of the early (2.5–3.5 h n = 4) and the late (3.5–4.5 h n = 3) blastula. (**P**) Max time projection (t = 2 min) of a 3.5 h blastula embryo imaged with a single Z-plane through the center (lateral position). Arrowheads point to the hyperpolarized cells only appearing within the superficial blastomere (Appendix A). A White dashed line indicated the EVL region of the embryo. (**Q**) Max time projection (t = 3 min) of a 3.5 h blastula embryo imaged with a single Z-plane through the center (animal pole position). Arrowheads point to the hyperpolarized cells only appearing within the superficial blastomere. The white segmented circle in the center of the blastula contains no hyperpolarized cells. Scale Bar= 250 µm. (**R**–**AA**) Early-stage blastula embryo (3 hpf) zoomed still-frame images from a time-lapse video (1 min 17-s total time, Appendix A). Red arrows indicate whole cells that were hyperpolarized. (**U**) The red arrow points to a strongly hyperpolarized cell. (**V**) The red arrow points to an adjacent cell that signaled 5 s later. (**W**) The red arrow points to a new adjacent cell signaled after another 5.5 s. This pattern continued, with the arrow in panel (**X**) pointing to another new adjacent cell from panel (**W**) This pattern finally dissipated with the earlier signaling cells fading. Eventually, the last signaling cell in panel (**X**) faded in (**AA**). Time (lower right corner), hours: minutes: seconds. Scale Bar= 50 µm.

**Figure 4 cells-11-03586-f004:**
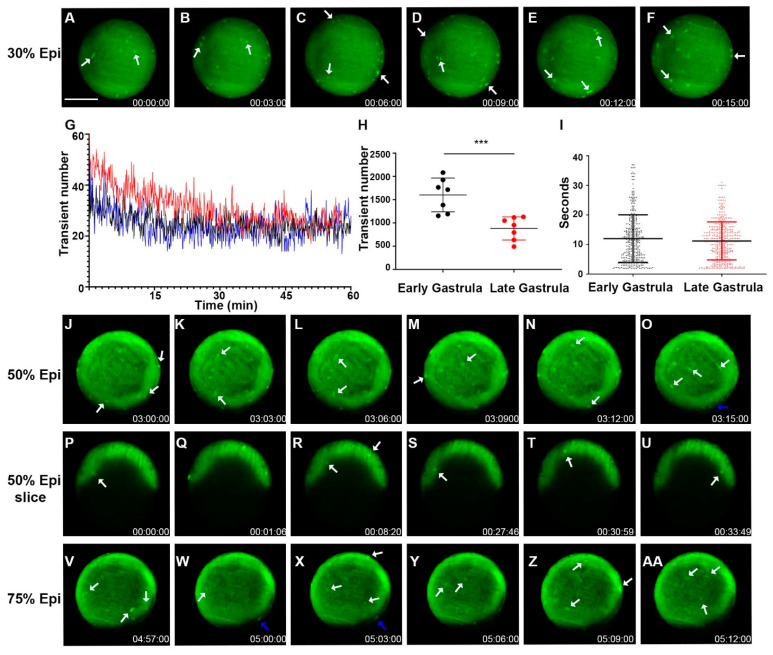
Zebrafish gastrulation exhibited whole-cell transient hyperpolarization in both superficial and deep cells. Early stage gastrula (30%) to 75% epiboly stages of the Tg (*ubi*: ASAP1) zebrafish embryo still-frame representative max-projection images from a time-lapse video (8 h total time, 3 min intervals, Appendix A). White arrows indicate whole cells that are hyperpolarized. (**A**–**F**) Early-stage gastrula embryo (~30% epiboly, animal pole view) showed whole-cell hyperpolarization in the EVL. (**G**) Average number of transients occurred at a given time point from a 60 min acquisition. The total number of hyperpolarized cells fluctuates over time. Each colored line indicates different fish embryos. (**H**) The total number of Vm transients occurred within the early (30% epiboly to shield) and late (shield-75% epiboly) gastrula embryo (n = 7 embryos for each group). Asterisks indicate a statistical significance of *p* < 0.001. (**I**) Vm transient duration of the early and late gastrula embryos (n = 4 embryos for each group). (**J**–**O**) Gastrula period embryos (50% epiboly) images from a time-lapse video (3 min intervals, Appendix A). Cell signals were seen in both the EVL (white arrows) and YSL (blue arrows). Overall signals were increased along the edge of the embryo where the embryonic shield was forming. (**P**–**U**) Time-lapse images of a 50% epiboly gastrula period embryo imaged with a single Z-plane through the center (lateral position). White arrows point to the hyperpolarized cells present within the deep cells (Appendix A). (**V**–**AA**) Gastrula period embryo 75% epiboly images from a time-lapse video (3 min intervals, Appendix A). Cell signals were seen in both the EVL (white arrows) and YSL (blue arrows). Overall signals were increased along the edge of the embryo where the embryonic shield was forming. Time (lower right corner), hours: minutes: seconds. Scale Bar= 250 µm.

**Figure 5 cells-11-03586-f005:**
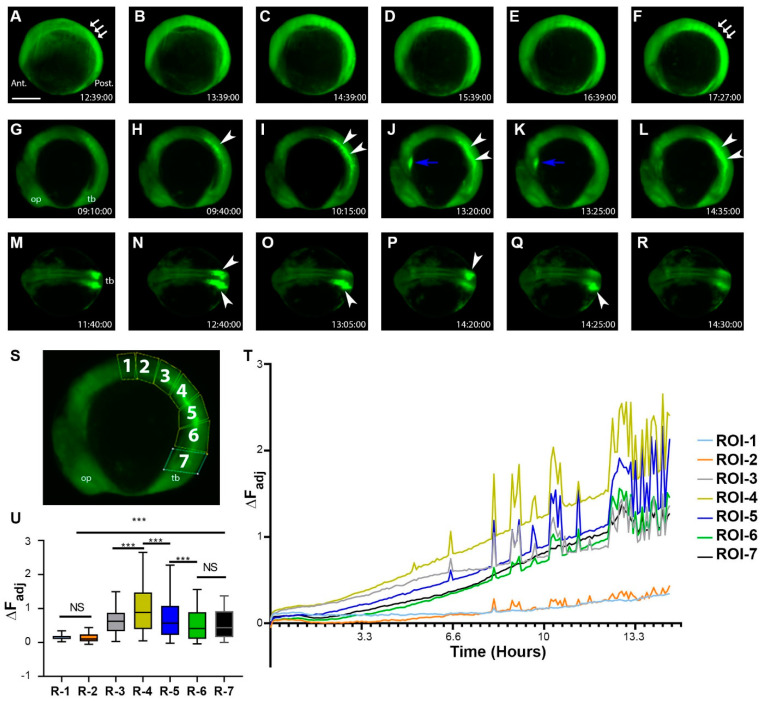
During the segment period, more complex and dynamic cellular bioelectric signals occurred at the tissue level. (**A**–**F**) Segmentation period (bud stage-6 somite stage, 1 h intervals, Appendix A). Somites and the posterior region of the embryo had an increased level of fluorescence. White arrows point to the somites. Note the relatively low fluorescent signals within the head region. (**G**–**L**) Left lateral time-lapse images of 10–16 somite zebrafish embryos (Appendix A). White arrowsheads point to the strong hyperpolarization of somites. Blue arrows point to Vm signals in the developing heart. (**M**–**R**) Dorsal view time-lapse images of 10–16 somite zebrafish embryos. White arrowheads indicate somite regions with strong hyperpolarization (Appendix A). (**S**) Embryo with positions of ROIs (1–7) used to calculate mean fluorescence and corresponding ∆F_Adj_. (**T**) ∆F_Adj_ over time of ROIs in panel (**S**). All colored lines represent the change in fluorescence intensity of the designated ROI at each time point. Signals appeared to increase over time as somites became more developed. The number of fluctuations also increased as more somites were generated. (**U**) The mean ∆F_Adj_ for each ROI for the entire duration of the time-lapse video. ROIs 1–2 showed the least amount of activity (most anterior somites), ROIs 3–5 showed the most activity (middle age somites), and ROIs 6–7 showed a moderate amount of activity (youngest somites/presomitic mesoderm/tailbud region). Asterisks indicate a statistical significance of *p* < 0.001. NS, not statistically significant. Scale Bar= 250 µm.

**Figure 6 cells-11-03586-f006:**
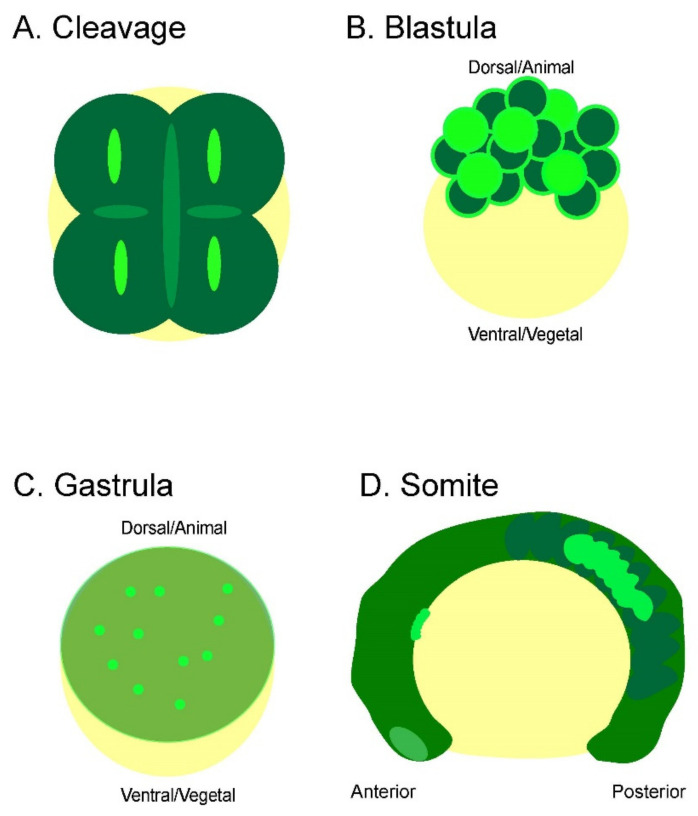
Summary of bioelectric signaling during zebrafish embryogenesis. Each early embryonic zebrafish developmental period has distinct yet overlapping bioelectricity signals and/or patterns. (**A**) The embryonic cleavage period is marked by cleavage furrow-associated Vm fluctuations that precede and persist cytokinesis. These signals become less synchronized and stable, starting around the 16-cell stage. (**B)** Whole-cell transient Vm signals characterize the blastula period. However, these signals are restricted to the superficial blastomere and are not seen within the deeper cells at this stage. In addition, intercellular signaling can be observed between adjacent cells. (**C**) The gastrulation period continues to display whole-cell transient hyperpolarization within the superficial blastomere and begins to occur within the deeper cells during epiboly. (**D**) Strong Vm transient signals mark the somite period. These signals can be whole or partial somites and are either unilateral or bilateral. The signals are more concentrated in the middle and posterior somites (bright green highlights hyperpolarization).

## Data Availability

Data is contained within the article or Appendix A.

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
