# Peer review of "Zebrafish Embryos Display Characteristic Bioelectric Signals during Early Development"

_cells, 2022, doi:10.3390/cells11223586_

Round 1
Reviewer 1 Report
In this MS, Silic et al. systematically analyzed changes in membrane hyperpolarization during zebrafish early development using Tg ASAP1 fish, a reporter fish line expressing GEVI under an ubiquitous promoter. This Tg fish line was reported in 2018 by this group but the signal too weak to be analyzed by conventional microscopy. In this study, the authors applied light sheet microscopy (LSM), which has enhanced sensitivity, speed, signal-to-noise ratio. The authors observed several patterns. During the cleavage period, embryos show cleavage furrow-associated Vm fluctuations that precede and persist cytokinesis. During the blastula stage and gastrulation stages, whole cell transient Vm signals are observed in the superficial blastomere but not seen within the deeper cells. Later in development, strong Vm transient signals are seen in some somites. These observations are new and potentially important. The MS is well prepared and easy to follow.
This reviewer only have a few minor comments:
1) This is a descriptive study in nature. This is fine but it would be nice to add some biological experiments. For instance, is it possible to cross this line with their long-fin mutant fish line discussed in line 355-357? Alternatively, a simple mRNA injection experiment can be performed to overexpress some of the kcnma channels to see if they can alter the Vm signal?
2) Is it possible to add some data to show the observed signal indeed reports Vm changes? For example, adding K channel blockers or depolarizing drugs?
Author Response
1) This is a descriptive study in nature. This is fine but it would be nice to add some biological experiments. For instance, is it possible to cross this line with their long-fin mutant fish line discussed in line 355-357? Alternatively, a simple mRNA injection experiment can be performed to overexpress some of the kcnma channels to see if they can alter the Vm signal?
Response:
We agree that adding more functional evidence will be great. However, this will take some effort. We have tried to image one of our long-fin mutant fish embryos but have not identified a time window with a clear difference. This will be one of our focuses in the long-fin project in the future. The mRNA injection of kcnma will destroy the integrity of the fish embryo. Thus, it could lead to abnormal Vm changes and large diversity of phenotypes. We have observed some damaged fish embryos showed uncharacteristic oscillating Vm signals at the cleavage furrows, and constant hyperplorized cells at later stages.
2) Is it possible to add some data to show the observed signal indeed reports Vm changes? For example, adding K channel blockers or depolarizing drugs?
Response:
The voltage reported by ASAP1 has been extensively validated in the human cells, mouse and fly brains, and even zebrafish brains(Nat Neurosci. 2014;17(6):884-9 Biophysical journal. 2017; 113(10): 2178-2181. eLife. 2017; 6:e25690. Sci Rep. 2018; 8, 6048. ). We have a parallel project on manipulating Vm using the chemogenetic tool, DREADD, and we confirmed that DREADD could modulate Vm reported by ASAP2s in zebrafish melanocytes (https://www.biorxiv.org/content/10.1101/2021.06.22.449481v1.full). Furthermore, in our JOVE paper (J Vis Exp 2018, e57330), the ASAP1 signal changed (neuromuscular firing) when the fish larva moved/twisted. This suggested that the ASAP1 signal indeed reports Vm changes.
To incorporate the above two suggestions, we added one sentence at the end of the manuscript (Lines 394-396). “In the future, the biological roles of embryonic Vm could be further examined with zebrafish ion channel mutants, newly developed optogenetic, or chemogenetic tools such as DREADD or uPSAM [54-56].”
Reviewer 2 Report
Silic et al in this manuscript used a voltage indicator reporter line to investigate cell membrane potential signals during multiple stages of zebrafish embryogenesis, including cleavage, blastula, gastrula and segmentation stages. They were able to quantify microscopy images, and identify temporal and spatial changes of cell membrane potential during development.
The manuscript was well written, data were clearly presented, and conclusions were supported by the experimental results. This work provides in vivo voltage signal patterns during embryogenesis, and would be useful to guide follow-up studies focusing on molecular mechanism of bioelectric signaling.
Minor suggestions to further emphasize the significance of this study:
1.Add more information in the introduction discussing advantages of genetic biosensors over chemical dyes/probes
2.Could the authors add more discussion about potential functional role of stage-specific bioelectric signal patterns during zebrafish embryogenesis?
Author Response
- Add more information in the Introduction discussing advantages of genetic biosensors over chemical dyes/probes
Response:
We added the information in the Introduction, lines 68-71.
“Genetically encoded biosensors generally allow sensitive and real-time dynamic assays for monitoring cells under natural physiological conditions. While chemical dyes/probes usually have limited lifetimes, relatively slow response, and delivery challenges due to tissue specificity and penetration.”
- Could the authors add more Discussion about potential functional role of stage-specific bioelectric signal patterns during zebrafish embryogenesis?
Response:
The biological function of embryonic bioelectric signals remains largely unknown. We added a new paragraph to the Discussion, Lines 365-372.
“ The functions of these unique developmental stage-specific bioelectric signal patterns during zebrafish embryogenesis remain largely unexplored. They could be related to cell cycle or cytokinesis, as previously suggested by ion channel studies from multiple species [39,40]. As most Vm transients were found in the peripheral regions during the blastula and gastrula stages, they likely play instructional roles in cell growth, differentiation, and organ patterning. As electric signals are correlated with calcium signals in neural tissues, it is also possible that the electric transients are just a reflection of calcium signal alterations in certain tissues, although this possibility is not high. Another possibility is the opposite, the electric signals trigger calcium signal.”